# Growth Restriction and Genomic Imprinting-Overlapping Phenotypes Support the Concept of an Imprinting Network

**DOI:** 10.3390/genes12040585

**Published:** 2021-04-17

**Authors:** Thomas Eggermann, Justin H. Davies, Maithé Tauber, Erica van den Akker, Anita Hokken-Koelega, Gudmundur Johansson, Irène Netchine

**Affiliations:** 1Institute of Human Genetics, Medical Faculty, RWTH Aachen University, 52062 Aachen, Germany; 2Department of Paediatric Endocrinology, University Hospital Southampton, Southampton SO16 6YD, UK; Justin.Davies@uhs.nhs.uk; 3Research centre of rare diseases PRADORT, Childrens Hospital, CHU Toulouse, Toulouse Institute of Infectious and Inflammatory Diseases (Infinity), INSERM UMR1291-CNRS UMR5051-Tolouse III University, 31062 Toulouse, France; tauber.mt@chu-toulouse.fr; 4Erasmus University Medical Center, University Medical Center Rotterdam, 3015 GD Rotterdam, The Netherlands; e.l.t.vandenakker@erasmusmc.nl; 5Erasmus University Medical Center, Pediatrics, Subdivision of Endocrinology, 3015 GD Rotterdam, The Netherlands; a.hokken@erasmusmc.nl; 6Department of Internal Medicine and Clinical Nutrition, Institute of Medicine, Sahlgrenska Academy, University of Gothenburg and Department of Endocrinology, Sahlgrenska University Hospital, 413 45 Gothenburg, Sweden; gudmundur.johannsson@medic.gu.se; 7Medical Faculty, AP-HP, Armand Trousseau Hospital-Functional Endocrine Research Unit, INSERM, Research Centre Saint-Antoine, Sorbonne University, 75012 Paris, France; irene.netchine@aphp.fr

**Keywords:** imprinting disorders, growth restriction, overgrowth, differentially methylated regions, imprinted gene network, Silver-Russell syndrome, transient neonatal diabetes, Prader-Willi syndrome, temple syndrome, pseudoparahypoparathyreoidism

## Abstract

Intrauterine and postnatal growth disturbances are major clinical features of imprinting disorders, a molecularly defined group of congenital syndromes caused by molecular alterations affecting parentally imprinted genes. These genes are expressed monoallelically and in a parent-of-origin manner, and they have an impact on human growth and development. In fact, several genes with an exclusive expression from the paternal allele have been shown to promote foetal growth, whereas maternally expressed genes suppress it. The evolution of this correlation might be explained by the different interests of the maternal and paternal genomes, aiming for the conservation of maternal resources for multiple offspring versus extracting maximal maternal resources. Since not all imprinted genes in higher mammals show the same imprinting pattern in different species, the findings from animal models are not always transferable to human. Therefore, human imprinting disorders might serve as models to understand the complex regulation and interaction of imprinted loci. This knowledge is a prerequisite for the development of precise diagnostic tools and therapeutic strategies for patients affected by imprinting disorders. In this review we will specifically overview the current knowledge on imprinting disorders associated with growth retardation, and its increasing relevance in a personalised medicine direction and the need for a multidisciplinary therapeutic approach.

## 1. Introduction

The epigenetic phenomenon of genomic imprinting allows the parent-of-origin-specific expression of only one allele of a gene in a spatial and temporal manner. From comprehensive studies in mice, numerous imprinted genes and their biological functions were identified [1], and syntenic imprinted chromosomal regions have been determined in humans though their regulation is dynamic and variable between different species [2]. The proper setting of imprinting marks is crucial for normal developmental processes in mammals, as genomic imprinting plays a key role in placental development, and particularly in intrauterine and postnatal growth. Disturbed imprinting patterns can therefore be associated with placental lesions in some disorders [3]. Additionally, its significance for metabolism, behaviour, cognitive development and function of the nervous system has also been proven (for review: [4,5]). In particular, in this review we will summarise the current knowledge on imprinting disorders with a focus on growth disturbance as a major clinical hallmark, and its increasing relevance in a personalised medicine direction.

## 2. Imprinting Disorders

It is therefore not surprising that the disturbance of the fine-tuned expression of imprinted genes in humans cause congenital disorders (so-called imprinting disorders; Table 1 and Table 2). The thirteen currently known congenital imprinting disorders share similar clinical features from the same spectrum, and they therefore overlap clinically as well as molecularly (for review: [6]) (Table 2). In accordance with the central physiological role of the involved imprinted genes in human ontogenesis, imprinting disorders exhibit clinical pictures comprising disturbances of growth, metabolism, and food-intake balance, pubertal timing and/or cognitive impairment. Additionally, tumour predisposition has also to be considered in at least two of them (i.e., Beckwith–Wiedemann syndrome/BWS, Kagami–Ogata syndrome/KOS14).

Imprinting regulation is based on the covalent modification of nucleotides (i.e., 5-methylcytosine, 5mC), posttranslational modifications of histone proteins, noncoding RNAs and chromatin confirmation. Imprinted genes cluster at specific chromosomal regions, called differentially methylated regions (DMRs), which control their expression (an example is shown in Figure 1). In imprinting disorders, four different types of molecular alterations can occur (for review: [6]), but with different frequencies (as examples listed in Table 1 for those entities characterised by growth restriction) whereas (i) single nucleotide variants (SNVs), (ii) copy number variants (CNVs) and (iii) uniparental disomy (UPD) of one of the two paternal alleles represent alterations of the DNA itself; (iv) epimutations commonly comprise altered methylations of 5mC, either by loss or gain of methylation (LOM, GOM). However, all four molecular changes are assumed to be associated with the changed expression of imprinted genes in humans, resulting either in an increased or decreased expression.

Whereas SNVs, CNVs and UPDs belong to the spectrum of naturally occurring mutations, the basic mechanisms behind epimutations are only partly understood. They comprise both exogeneous factors (such as assisted reproductive technologies) as well as genetic determinants with an impact on the proper establishment, maintenance and erasure of imprinting marks [16]. In a subgroup of patients with epimutations, a broad range of altered imprinting methylation patterns can be observed, affecting multiple imprinted loci (multilocus imprinting disturbances, MLID, see below).

## 3. Factors Causing Aberrant Imprinting 

Intrauterine growth and development are influenced by both maternal and foetal determinants, some of them with long-term consequences in later life (for review: [17]). Maternal determinants comprise environmental factors as well as monogenic predispositions. On the foetal side, chromosomal as well as monogenic causes can severely affect the foetal constitution. However, the transition between environmental and genetic causes for prenatal pathologies is fluent, and altered imprinting marks play a major role.

### 3.1. Maternal Determinants with an Impact on Imprinting

Due to the complex processes of the erasure, establishment and maintenance of imprinting in gamete formation, fertilisation and early embryonic development, these stages are particularly vulnerable to environmental influences, and it is not surprising that maternal nutritional status and intrauterine exposure to chemical pollutants have been identified to influence proper genomic imprinting (e.g., [18,19]). Accordingly, assisted reproductive technologies (ART) might also affect genomic imprinting (for review: [20]) and have been suggested as risk factor for imprinting disorders. Maternal genetic factors with an impact on the maintenance of imprinting marks have recently been discovered, and these maternal affect mutations have a impact on the genes encoding the proteins of the subcortical maternal complex (SCMC) which maintains the imprinting marks in the early embryo (see below). Accordingly, genetic variants in SCMC proteins (NLRP2, NLRP5, NLRP7, PADI6, KHDC3L, OOEP) have been suggested to alter the imprinting status of the oocyte and the early embryo, resulting in MLID in the offspring (for review: [21]). In general, the pregnancies of women carrying maternal affect mutations can have different courses, ranging from hydatidiform moles and miscarriages to birth of children with the clinical features of various imprinting disorders and aneuploidies (for review [22]).

### 3.2. Foetal Determinants with an Impact on Imprinting

Molecular alterations of the foetus might have a severe impact on foetal viability and health status. In addition to the well-known pathogenic nature of chromosomal aberrations and monogenic mutations (e.g., trisomies, *FGFR3* mutations), disturbed genomic imprinting is associated with a broad spectrum of clinical features, some of them manifesting intrautero and often affecting growth (Table 1). As described earlier, the disturbed imprinting marks can be caused by maternal determinants, but alterations in the foetal genome can also cause aberrant imprinting. These foetal genomic variants comprise both cis- and trans-acting factors (for review: [16]). Among others, cisacting factors comprise transcripts from imprinted regions which are needed to establish or maintain methylation across the respective DMR, like the *KCNQ1* transcript with an impact on the *KCNQ1OT1*:TSS-DMR in 11p15.5 [23]. Another example of cisfunctioning alterations are defective target sites for binding proteins (e.g., CTCF and OCT4/SOX2) which protects the DMR from methylation (e.g., [24,25]).

Trans-acting factors causing aberrant imprinting are localised outside the imprinting region, and include pathogenic variants of factors mediating the proper establishment or maintenance of the imprinting methylation. An example are mutations in *ZFP57*, a gene expressed from the foetal genome in the embryo that that protects specific DMRs from demethylation during early embryogenesis [26]. In fact, the aforementioned SCMC genes can also be assigned to the group of trans-acting factors.

### 3.3. The Imprinted Genes Network/IGN

Several studies in higher mammals indicate that imprinted genes are coregulated or that some genes have an impact on the expression of others, therefore a network of imprinted genes (IGN) has been suggested [27,28,29]. The disturbance of one of these interacting genes not only affects the expression of the factor itself, but might also deregulate functionally related genes, including both imprinted and non-imprinted genes [30]. The clinical consequence of these physiological interactions is the clinical overlap between some of the imprinting disorders, e.g., between Silver-Russell Syndrome (SRS) and Temple Syndrome (TS14) (Figure 2). In SRS, the disruption of the paternal imprinting pattern on chromosome 11 diminishes the production of IGF II, a major controller of foetal growth. The disruption of paternal imprinting on chromosome 14 in TS14 modifies the expression of the *IGF2* gene, even though its paternal imprinting pattern on chromosome 11 is correct. TS14 patients have increased expression of *MEG3*, *MEG8*, two maternally expressed noncoding long RNA as well as increased miRNAs which might modify the IGN and in particular *IGF2* expression [30]. This example illustrates the close interactions between genes, the expression of which is regulated by genomic imprinting. However, the regulation mechanisms and interactions of the majority of the imprinted loci are far from being understood.

## 4. Growth Restriction and Imprinting Disorders

The observation that intrauterine and postnatal growth disturbances are dominant clinical features of imprinting disorders fits in with the role of many imprinted genes in placental function and foetal growth (for review: [31]). In fact, several genes with an exclusive expression from the paternal allele have been shown to promote foetal growth in mice (e.g., *Igf2, Dlk1, Peg1*), whereas maternally expressed genes like *Grb10* and *CDKN1C* suppress it. The evolution of this correlation has been explained by the “parental conflict hypothesis” which suggests different interests of the maternal and paternal genomes, aiming for the conservation of maternal resources for multiple litters versus extracting maximal maternal resources [32]. This hypothesis is not only based on the correlation between the imprinting regulation of specific genes and their immediate role in growth as growth promotors or inhibitors, but also on the observation that genes with an impact on maternal care (*Mest/Peg1*), pup behaviour (*Magel2, XLas*) and thermogenesis (*Dlk1, Grb10, Cdkn1c*) as essential prerequisites for neonatal survival in mice are imprinted (for review: [2,31]). As not all imprinted genes in higher mammals show the same imprinting pattern in different species, the findings from animal models are not always transferable to the status and role of imprinted genes and clusters in human [2]. However, these discrepant as well as overlapping imprinting patterns in different species help to understand the role of imprinting during evolution and its contribution to mediate similar but also different functions of the same factor, as shown for Grb10/GRB10 [33].

In humans, the role of imprinted genes and clusters has mainly indirectly been delineated as molecular alterations (CNVs, UPDs, epimutations) in these regions, associated with growth disturbances. However, in two imprinted genes, pathogenic variants causing growth restriction or overgrowth have already been identified to. These variants in *IGF2* and *CDKN1C* are inherited in an autosomal-dominant mode, but according to the imprinting status of the genes, the growth disturbance phenotypes only occur in case of paternal (*IGF2*, 11p15.5) or maternal (*CDKN1C*) inheritance. The pathogenic variants of the paternally expressed *IGF2* gene are associated with growth retardation and the SRS phenotype [34]. In contrast, loss-of-function mutations in the maternally expressed *CDKN1C* gene cause overgrowth, whereas gain-of-function variants have been described in growth restriction phenotypes (for review: [35]). For both factors, the contribution to human growth is obvious, either as a (prenatal) growth factor (IGF II) or a negative regulator of cell proliferation. As these genes are members of the growth pathways, it is not surprising that mutations in further non-imprinted factors of these members cause similar phenotypes [36].

## 5. Imprinting Disorders Associated with Growth Restriction

As already described, growth disturbance is a major feature of many imprinting disorders, including overgrowth in BWS and KOS14, and growth restriction in transient diabetes mellitus (TNDM), SRS, TS14, PWS, and the chromosome 20 associated disorders (Table 2). To address the topic of this overview, these entities are briefly summarised in the order of their chromosomal localization, with regard to the characteristic symptoms and the common clinical management to allow a quick overview (see Table 1). However, the symptoms of imprinting disorders often overlap; therefore, a specific clinical diagnosis is not always possible.

## 6. Transient Neonatal Diabetes Mellitus 1 (TNDM 1; 6q24)

### 6.1. Molecular Characteristics

TNDM1 can be caused by different molecular mechanisms linked to the 6q24 region: Partial or complete paternal UPD 6, duplication of the paternal allele at 6q24, and loss of methylation (LOM) at the maternal *PLAGL1*:alt-TSS-DMR. All these mechanisms lead to the overexpression of *PLAGL1* and *HYMAI*. *PLAGL1* encodes a zinc finger protein and regulates PACAP1 that has a key role in stimulating insulin secretion by pancreatic β cells. The overexpression of *PLAG1* may reduce the number of the pancreatic β cells and diminished insulin secretion (for review: [37]) LOM of the *PLAGL1*: alt-TSS-DMR can be either isolated or associated with MLID due to recessive loss of function *ZFP57* mutations in almost half of the cases [38].

### 6.2. Clinical Characteristics, Diagnosis and Therapy

TNDM1 is characterised by intrauterine growth retardation and hyperglycaemia without ketoacidosis during the neonatal period. The diabetes mellitus develops during the first weeks of life, before 3 months in 100% of the cases, with a remission by the age of 18 months. About fifty percent of the patients with TNDM are at risk of type 2 diabetes during adolescence or early adulthood or are at risk of diabetes mellitus during pregnancy. Macroglossia affects up to 50% of the infants with TNDM1 and about 20% may also have a minor anomaly of the abdominal wall. Cardiac malformations, renal and urinary malformations, nonautoimmune anaemia, hypothyroidism with gland in situ and neurological disorders may also be associated. Insulin-based treatment is necessary but is difficult to manage due to the low birth weight [39].

## 7. Silver-Russell Syndrome (SRS; 7p12, 7q32, 11p15.5)

### 7.1. Molecular Characteristics

Among patients with a positive SRS clinical diagnosis, a molecular anomaly can be identified in about 60% of the patients. The main molecular causes are LOM of the distal imprinting control region (*H19/IGF2*:IG-DMR) on 11p15.5 (50%) and maternal uniparental disomy of chromosome 7 (5–10%). Other rare 11p15.5-related molecular defects, such as maternal duplications, *CDKN1C* and *IGF2* point mutations affecting the maternal allele and the paternal allele, respectively, have also been implicated in SRS, as well as *HMGA2* and *PLAG1* mutation or deletion, two genes which form part of a pathway with *IGF2 (28)*. In about 30–40% of patients with a clinical diagnosis of SRS, the molecular aetiology of the clinical features remains unknown, probably due to other molecular mechanisms [9].

### 7.2. Clinical Diagnosis and Therapy

SRS is a rare but well-recognised imprinting disorder with prenatal and postnatal growth retardation. Currently the clinical diagnosis is based on a combination of the characteristic features evaluated by a clinical scoring system (Netchine-Harbison Clinical Scoring System [40]. Relative macrocephaly at birth is a key criterion for this diagnosis and exposes the patient to a high risk of hypoglycaemia, which should be carefully monitored [9]. The first Consensus Statement on Silver-Russell Syndrome was held in 2015 [9]. Considerable overlap exists between the care of individuals born small for gestational age and those with SRS. However, many specific management issues exist and evidence from controlled trials remains limited. The management of children with SRS requires an experienced, multidisciplinary approach. Specific issues include growth failure, severe feeding difficulties in early childhood, gastrointestinal problems, hypoglycaemia, body asymmetry, scoliosis, motor and speech delay, sleep apnoea and psychosocial challenges. An early emphasis on adequate nutritional status is important, with awareness that rapid postnatal weight gain might lead to subsequent increased risk of metabolic disorders. The benefits of treating SRS patients with growth hormone (rGH) include improved body composition, motor development and appetite, reduced risk of hypoglycaemia and increased height [41,42]. Clinicians should be aware of possible premature adrenarche, fairly early and rapid central puberty and insulin resistance. Treatment with gonadotropin-releasing hormone analogues can delay the progression of central puberty and preserve adult height potential but requires further studies. Long-term follow up is essential to determine the natural history and optimal management in adulthood (see transition section).

## 8. Temple Syndrome (TS14; 14q32), (Central Precocious Puberty, CPPB) 

### 8.1. Molecular Characteristics

Temple syndrome (TS14) can result from maternal UPD of chromosome 14 (~29% of cases), paternal deletion within the 14q32 imprinting region (*~11% of cases) and paternal hypomethylation of the intergenic MEG3/DLK1 IG-DMR (~60% of cases) (for review: [43]). A smaller paternal deletion in 14q32 only affecting the *DLK1* gene causes precocious puberty, which is a common feature of TS but is not causing foetal growth restriction.

### 8.2. Clinical Diagnosis and Therapy

The phenotype of TS14 overlaps with both SRS and PWS. TS14 is characterised by pre- and postnatal growth failure although this is not as severe as in SRS. Approximately 50% of TS will have relative macrocephaly at birth [10]. IUGR and SGA may be present in up to 75% of cases, similar to SRS. Hypotonia is a prominent feature (68–83%) [10,11], but unlike PWS, it is not profound during the neonatal period but does persist into childhood. Early onset puberty is typical (86%) and often requires intervention with gonadotropin-releasing hormone analogues and obesity is also very frequent (49%) [41]. Treatment with rGH has been shown to have a short-term beneficial effect on linear growth [44]. It is unclear whether there is a definite association with thyroid cancer and TS14 caused by 14q32 deletion [45] but this should be monitored for patients with TS14 caused by 14q32 deletion by regular thyroid ultrasounds. Multidisciplinary care is necessary for patients with TS14.

## 9. Prader-Willi Syndrome (PWS, 15q11q13)

### 9.1. Molecular Characteristics

The 15q11q13 region harbours a DMR (*SNURF*:TSS-DMR) which is paternally unmethylated and maternally methylated [46]. This results in differential expression of following genes in the region: *NDN*, *MAGEL2*, *MKRN3*, *NPAP1*, *SNURF-SNRPN* and several snoRNAs are paternally expressed whereas *UBE3A* is maternally expressed in the brain only. The molecular causes of PWS (and Angelman syndrome) are a 5-7 Mb deletion in 15q11q13, a UPD or an imprinting defect which is mostly sporadic without a DNA sequence change [47,48]. However, in 10–15% of all imprinting defect cases the defect is due to an IC-deletion in 15q11.2 [49]. Rare cases of microdeletions of the *SNORD116* gene locus have been published leading to PWS-like phenotype.

### 9.2. Clinical Diagnosis and Therapy

Prader-Willi syndrome is a complex neurodevelopmental multisystemic disorder (see also genereviews: NBK1330), characterised by a characteristic trajectory from severe neonatal muscular hypotonia, poor desire to sit and failure to thrive to insatiable appetite and hyperphagia which will lead to early extreme obesity when food intake is not restricted, short stature and hypogonadism, as well as variably impaired cognitive function, distinct behavioural problems and psychiatric comorbidities (e.g., psychosis and autism-spectrum disorders) [12,50]. Hypothalamic dysfunction accounts for many clinical aspects of the PWS-phenotype [51]. The prevalence of scoliosis in PWS is 80% above the age of 10 years [52,53].

The complexity of PWS requires a multidisciplinary therapeutic approach. Growth hormone therapy is approved for children with PWS and is recommended to ameliorate body composition by decreasing fat mass and increasing lean body mass, not only to improve adult height [54]. Other endocrine manifestations requiring hormone replacement therapy are hypogonadism in almost all, as well as central hypothyroidism and adrenal insufficiency in some PWS-individuals [55]. Morbid obesity and related complications must be avoided by dietary measures and strict control of food intake. Physiotherapy and psychological support are essential—some patients also need psychiatric treatment, most often as adolescents and adults. 

## 10. Pseudoparahypoparathyreoidism/Inactivating PTH/PTHrP Signalling Disorders (20q13.32)

### 10.1. Molecular Characteristics

Pseudohypoparathyroidism (PHP) comprises a heterogeneous group of rare related disorders characterised by diminished activation of the Gsα/cAMP/PKA signalling pathway by the parathyroid hormone (PTH) and other hormones (for review: [56]). Gsα is encoded by the imprinted gene *GNAS* (20q13.3). This gene is biallelically expressed in most tissues, but mostly maternally expressed in thyroid, renal proximal tubule, pituitary and ovary tissues. This tissue-specific monoallelic expression of Gsα explains most of the clinical outcomes that depend on the parental origin of the *GNAS* mutation. *GNAS* encodes multiple transcripts using different promoters that have a parent-specific methylation. PHP1A is caused by inactivating DNA variants on the *GNAS* maternal allele, whereas variants on the paternal allele are mainly associated with PseudoPHP (PPHP). PHP1C is considered to be a variant of PHP1A. PHP1B is secondary to abnormal methylation in the Differential Methylated Regions (DMRs) associated with the *GNAS* complex locus. This abnormal methylation is either partial or complete and can affect one or multiple DMRs within 20q13.3. In 15–20%, these defects are familial, with an autosomal maternal dominant mode of inheritance (AD-PHP1B). In most sporadic cases of PHP1B, the methylation of two or more DMRs of this complex *GNAS* locus are also affected. For around 8–10% of these sporadic cases, the methylation defects are caused by paternal uniparental of the chromosomal region 20q13.3 (upd(20q13)pat). 

### 10.2. Clinical Diagnosis and Therapy

In very young children, the clinical signs are often unspecific (being born small for gestational age (SGA), early onset obesity before two years or transient hypothyroidism) and at this period the diagnosis is therefore difficult. Later on, growth failure, brachydactyly, ectopic ossification, obesity, and/or hypocalcemia leading to neuromuscular symptoms develop and may better orient this diagnosis. In the majority of patients with PHP, the most important clinical signs are symptoms of hypocalcaemia due to PTH resistance. The diagnosis of PHP should be based on major criteria, including resistance to PTH (hypocalcemia, hyperphosphatemia and elevated serum levels of PTH in the absence of vitamin D deficiency, abnormal magnesium levels, and/or renal insufficiency), ectopic ossifications, brachydactyly and early onset obesity [13]. Patients should be screened at diagnosis and during follow-up for PTH resistance, TSH resistance, growth hormone deficiency, hypogonadism, skeletal deformities, oral health, weight gain, glucose intolerance or type 2 diabetes and hypertension, as well as subcutaneous and/or deeper ectopic ossifications and neurocognitive impairment. A multidisciplinary team is necessary for the diagnosis, follow-up and to propose treatments for all these clinical outcomes. Severe symptomatic hypocalcemia should be treated to target levels of calcium and phosphorus within the normal range while avoiding hypercalciuria. Physical therapy and skin care are critical for the prevention of complications due to ectopic ossifications. Regular monitoring of growth, skeletal maturation and GH secretion should start around the age of three years. Patients born SGA without catch-up growth or patients with a GH deficiency should be considered for rhGH treatment. BMI and eating behaviour should be regularly monitored, psychological support and educational programs should be proposed. A multidisciplinary therapeutic approach is therefore necessary.

## 11. Mulchandani–Bhoj–Conlin Syndrome (MBCS; 20q11q13)

### 11.1. Molecular Characteristics

Maternal UPD of chromosome 20 (upd(20)mat) has recently been named Mulchandani–Bhoj–Conlin syndrome (MBCS) [14]. Up to now, UPD is the only molecular alteration in these patients (see below).

### 11.2. Clinical Diagnosis and Therapy

The phenotype of MBCS is rather unspecific with prenatal growth retardation, short stature with proportional head circumference, and feeding difficulties as the major features [14,57]. Neurodevelopment seems to be normal. Due to the lack of specific features, upd(20)mat as the currently only molecular alteration of MBCS has mainly been identified in patients referred for SRS testing. Data from growth hormone treatment in a small number of cases does not indicate harm for the patients; therefore the clinical management might lean on that for SRS [9].

## 12. Upd(6)mat and Upd(16)mat: Further Imprinting Disorders?

With maternal UPDs of chromosome 6 and 16, two further genetic constitutions putatively associated with genomic imprinting have been reported in growth retarded patients. However, for both molecular conditions it is discussed whether the clinical features might be caused or modified by the presence of a trisomic cell line, as UPD can be associated with trisomy mosaicism due to its mode of formation. In principle, this possibility applies for all UPDs, but depending on the formation mechanism and the gene content of the affected region, trisomy mosaicism does not play a relevant role. In fact, for upd(16)mat, the clinical heterogeneity has been attributed to mosaic trisomy 16 cell lines [58], but the recent identification of a case with an isolated methylation defect [59] might indicate that at least some features might be linked to an imprinting defect. In case of upd(6)mat, the impact of trisomy 6 cell lines has been suggested [60] whereas further evidence for an imprinting effect has not yet been published.

Thus far, very little information is available regarding treatment, but in case of patients with SGA without catch-up growth, rGH treatment under the SGA indication is possible.

## 13. Translation and Transition

The significant improvement of molecular diagnostics in the recent years increases the diagnostic yield in patients with imprinting disorders, and rare diseases in general. Furthermore, it allows a more precise molecular diagnosis as the prerequisite for a targeted and personalised therapy, as well as for genetic counselling of the family. As described for PWS, SRS and TS14, the molecular confirmation of these disorders is the indication for rGH treatment, in these and other imprinting disorders, clinical management depends on the molecular diagnosis as well. Due to their clinical heterogeneity and overlap (Table 2), patients suffering from imprinting disorders often remain without a diagnosis or are misdiagnosed, with a severe impact on their treatment, additional burden and life-long consequences. Thus an early and comprehensive molecular diagnostic workup allows an earlier and therefore more effective medical intervention. This early diagnosis of a genetically based disorder supports the patients and their family in their self-determined planning of life as early as possible, gives them access to the accurate multidisciplinary therapeutic approach and specific support patients groups [61].

Nevertheless, it has to be borne in mind that molecular genetic testing can confirm a clinical diagnosis, but never exclude it, thereby leaving numerous patients without molecular diagnosis (e.g., up to 40% in SRS; Table 1). As it has been shown for patients referred for SRS testing, it can be relevant to identify the genetic cause in this “idiopathic” group, as some of these genetic alterations can affect tumour predispositions genes like the *BLM* gene, associated with Bloom syndrome [9,62]. The recent progress in identifying (epi)genetic causes in patients with rare diseases is mainly based on the rapid development and implementation of next generation sequencing, and in several cohorts of patients with rare diseases its power has been proven (e.g., [63]).

In addition to the benefit of an accurate molecular diagnosis for the patients, this diagnosis allows an accurate prognosis of recurrence risks for family planning as the basis of genetic counselling. Asymptomatic carriers of pathogenic variants can be identified, and prenatal testing might be offered, when appropriate.

As a result of this improvement of molecular diagnostics and personalised therapies, an increasing number of patients with previously unidentified disorders reach adulthood, and need support in the transition from family and paediatric care to health care in adults with a specific multidisciplinary health care plan for their disorder [64]. The latter situation is reflected by the lack on data on health issues and quality of life of elder patients with imprinting disorders. In fact, the first data indicate increased risks for metabolic and orthopaedic problems in adulthood in some imprinting disorders (e.g., SRS [65]), but systematic studies are missing. These surveys are urgently needed as they will help to evaluate the efficiency of therapies like rGH treatment in childhood and their long-term consequences during adulthood. In general, transition infrastructure is needed to support patients with imprinting disorders during adolescence, and to achieve knowledge on the disease-related health problems in adulthood and are being developed in centres of reference for rare disorders and at the EU level in networks like the European Reference Network on Rare Endocrine Conditions (ENDO-ERN).

## 14. Conclusions and Outlook

With the rapid development of omic approaches comprising genomic next generation sequencing, transcriptomics (RNAseq) as well as methylomic, future research approaches will allow comprehensive insights in the role of genomic imprinting in human growth and its disturbances. These technical formats are indispensable tools to identify new pathophysiological mechanisms of human disorders and improve diagnostic algorithms. However, the data assessment has to be embedded in interdisciplinary discussions using all available clinical and molecular information, to further understand genomic imprinting and its functional relevance. This understanding might finally help to develop targeted drugs for imprinting disorders, as suggested for Angelman [66], Silver-Russell and Beckwith–Wiedemann syndromes [67].

## Figures and Tables

**Figure 1 genes-12-00585-f001:**
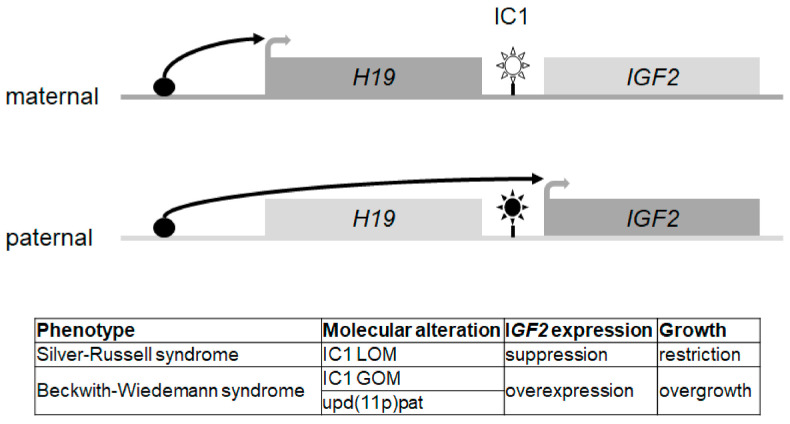
The imprinting centre 1 (IC1) region in 11p15 as an example for imprinting regulation and its disturbances. (LOM, loss of methylation; GOM, gain of methylation).

**Figure 2 genes-12-00585-f002:**
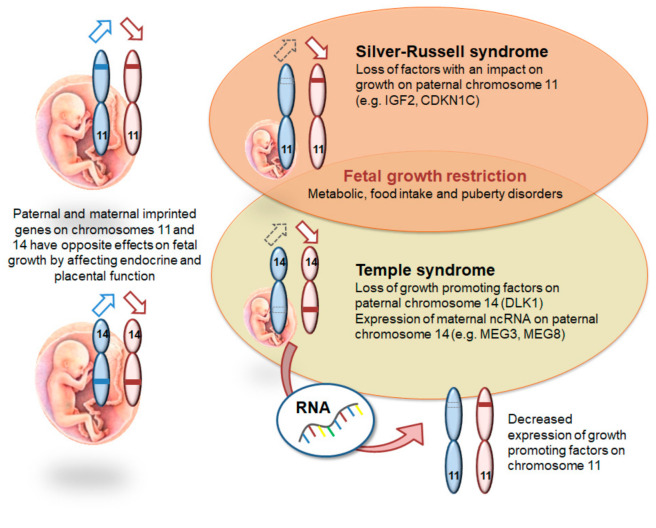
Interaction of imprinted genes on chromosomes 11 and 14 and (putative) functional relation between Silver-Russell and Temple syndromes.

**Table 1 genes-12-00585-t001:** Summary of the imprinting disorders characterised by growth disturbances, those disorders without altered growth are not listed. (GOM, gain of methylation; IUGR, intrauterine growth restriction; LOM, loss of methylation; PNGR, postnatal growth restriction; PTH, parathyroid hormone; UPD, uniparental disomy; MLID, Multilocus Imprinting Defects; for the nomenclature of DMRs see [7]); ? not yet reported or unclear).

Imprinting Disorder OMIM	Prevalence	Chromosome	Molecular Defect (Frequency)	MLID	Main Clinical Features	References
Transient neonatal diabetes mellitus (TNDM) 601410	1/300.000	6q24	- upd(6)pat: 41% - Paternal duplications: 29% - *PLAGL1:alt-TSS-DMR* LOM	30%	IUGR, transient diabetes mellitus, hyperglycaemia without ketoacidosis, macroglossia, abdominal wall defects	[8]
Silver-Russell syndrome (SRS) 180860	1/75.000–1/100.000	Chr 7 Chr 11p15	- upd(7)mat: 5–10% - upd(11p15)mat - 11p15 CNVs (<1%) - *H19*/IGF2:IG:DMR LOM: 30–60% *CDKN1C, IGF2, HMGA2, PLAG1* point mutations	1 case 7–10%	IUGR, PNGR, relative macrocephaly at birth, body asymmetry (11p15), prominent forehead, feeding difficulties in early childhood	[9]
Temple syndrome (TS14) 616222	unknown	Chr 14q32	- upd(14)mat: 29% - 14q32 paternal deletion: 10% - *MEG3/DLK1*:IG-DMR LOM: 61%	?	IUGR, PNGR, neonatal hypotonia, feeding difficulties in infancy, truncal obesity, scoliosis, precocious puberty, small feet and hands	[10,11]
Prader-Willi syndrome (PWS) 176270	1/25.000 –1/15.000	Chr 15q11.2	- 15q11–q13 paternal deletion: 55% - upd(15)mat: 42% - *SNURF*:TSS-DMR GOM: 2%	1 case?	Neonatal hypotonia, severe feeding difficulties in infancy (poor suck), hypogenitalism, PNGR, psychomotor developmental delay, intellectual disability, behavioural problems (tantrums), hyperphagia, (extreme) obesity, hypogonadism, hypopigmentation, scoliosis, abnormal pubertal progression small hands and feet	[12]
Pseudohypo-parathyroidism 1B (PHP1B) 603233	Unknown	Chr 20q13	- 20q13 maternal deletion: 8.5% - *GNAS* DMRs LOM: 42.5% - upd(20)pat: 2.5% - 20q13 point mutations: 46.5%	12.5%	macrosomia, PTH resistance, TSH resistance, Albright hereditary osteodystrophy, early onset obesity subcutaneous ossifications	[13]
Mulchandani-Bhoj-Conlin syndrome (MBCS) 617352	Unknown	Chr 20	- upd(20)mat		IUGR, PNGR, microcephaly, feeding difficulties, psychomotor developmental delay in some children	[14,15]

**Table 2 genes-12-00585-t002:** Overview on the clinical overlap between the currently known 13 imprinting disorders. It should be noted that only overlapping features are listed, independent of their frequencies.

Imprinting Disorder	OMIM	Chromosome	Imprinted Genes in the Region ^a^	IUGR	PNGR	Foetal Macrosomia	Postnatal Overgrowth	(Relative) Macrocephaly	Asymmetry	Abdominal Wall Defects	Macroglossia	Metabolic Disturbance	Feeding Difficulties in Infancy	(Truncal) Obesity	Scoliosis	Early or Precocious Puberty	(Neonatal) Hypotonia	Ccognitive Impairment	Embryonal Tumor	Placental Pathology	Polyhydramnios
Transient neonatal diabetes mellitus (TNDM)	601410	6q24	*PLAGL1*	yes						yes	Yes	hyperglycemia without ketoacidosis									
Silver-Russell syndrome ^b^ (SRS)	180860	11p15.5	*IGF2, CDKN1C*	yes	yes			yes	yes			hypoglycaemia/ insulin resistance in young adults	yes	yes ^c^	yes	yes					
	618905	7p13q32	*GRB10, MEST*	yes	yes			yes	yes			hypoglycaemia	yes		yes	yes	yes	yes ^d^			
Birk-Barel syndrome	612292	8q24.3	*KCNK9*														yes	yes			
Beckwith–Wiedemann syndrome (BWS)	130650	11p15.5	*CDKN1C, IGF2, H19*			yes	yes		yes	yes	Yes	hyperinsulinism							yes	yes	yes
Temple syndrome (TS14)	616222	14q32	*DLK1*	yes	yes			yes	yes			insulin resistance	yes	yes	yes	yes	yes	yes			
Kagami–Ogata syndrome (KOS14)	608149	14q32	*DLK1*	yes	yes	yes				yes			yes					yes	yes	yes	yes
(familial) central precocious puberty (CBBP)	(DLK1: 176290)	14q32	*DLK1*													yes					
Prader-Willi syndrome (PWS)	176270	15q11.2	*SNRPN*	yes	yes			no	facial asymmetry in neoates			hypoinsulinemia and high insulin sensitivity	yes	yes		yes	yes	yes			
Angelman syndrome (AS)	105830	15q11.2	*SNRPN, UBE3A*												yes			yes			
Central precocious puberty 2 (CPPB2)	615356	15q11.2	*MKRN3*													yes					
Schaaf-Yang syndrome (SYS)	615547	15q11.2	*MAGEL2*		yes							hyperinsulinism			yes		yes	yes			
Pseudohypoparathyroidism 1B (PHP1b)	603233	20q13	*GNAS* complex	yes	yes	yes	yes						yes	yes				Yes			
Mulchandani–Bhoj–Conlin syndrome (MBCS)	617352	20		yes	yes								yes								

IUGR intrauterine growth restriction; PNGR postnatal growth restriction; ^a^ only genes harboring DMRs are listed; ^b^ only subgroups associated with imprinted loci are shown; ^c^ possible in adulthood; ^d^ autism spectrum disorders and dystonic myoclonia have been reported.

## Data Availability

Not applicable.

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
