# Peer review of "Growth Restriction and Genomic Imprinting-Overlapping Phenotypes Support the Concept of an Imprinting Network"

_genes, 2021, doi:10.3390/genes12040585_

Round 1

Reviewer 1 Report

The review is important to fixate on a current state of knowledge in the field of genomic imprinting. At the same time, there are several brilliant recent reviews in this area, including prepared by the authors of this manuscript. See for example, Monk D. et al. Genomic imprinting disorders: lessons on how genome, epigenome and environment interact // Nat Rev Genet 2019, PMID: 30647469; Tucci V. et al. Genomic imprinting and physiological processes in mammals // Cell 2019, PMID: 30794780. From this point of view, it would be great if authors can emphasize the aim and novelty of their review for the 2021 year in the abstract and introduction. There are also some minor comments to the manuscript:

  1. Figures 1 and 2 are mixed up in the text.
  2. All abbreviations must be deciphered, including TSS. 
  3. Lines 1 and 2 on Page 7: monogenetic should be changed by monogenic.
  4. Lines 19 and 20 on Page 7: it would be better to indicate all known proteins of the subcortical maternal complex.
  5. The last paragraph on Page 7: please explain, what do you mean - "in ZFP57, a gene expressed from the fetal genome in the oocyte"?

Author Response

Reviewer 1:

The review is important to fixate on a current state of knowledge in the field of genomic imprinting. At the same time, there are several brilliant recent reviews in this area, including prepared by the authors of this manuscript. See for example, Monk D. et al. Genomic imprinting disorders: lessons on how genome, epigenome and environment interact // Nat Rev Genet 2019, PMID: 30647469; Tucci V. et al. Genomic imprinting and physiological processes in mammals // Cell 2019, PMID: 30794780. From this point of view, it would be great if authors can emphasize the aim and novelty of their review for the 2021 year in the abstract and introduction.

ANSWER: We have added two sentences in both sections to emphasize the focus of our paper.

There are also some minor comments to the manuscript:

Figures 1 and 2 are mixed up in the text.

ANSWER: We are sorry for this mistake and have corrected it.

All abbreviations must be deciphered, including TSS. 

ANSWER: SGA has additionally been added to the list of abbreviations. For the nomenclature of DMRs (as asked for TSS), we now refer to the respective publication in the “list of abbreviations” and the legend of figure 2.

Lines 1 and 2 on Page 7: monogenetic should be changed by monogenic.

ANSWER: Done.

Lines 19 and 20 on Page 7: it would be better to indicate all known proteins of the subcortical maternal complex.

ANSWER: Done.

The last paragraph on Page 7: please explain, what do you mean - "in ZFP57, a gene expressed from the fetal genome in the oocyte"?

ANSWER: We have replaced “oocyte” by “embryo”.

Reviewer 2 Report

This review provides an overview of genomic imprinting syndromes which feature fetal growth restriction principally from a clinical perspective. There is a detailed presentation of the molecular alterations reported in specific syndromes and highlighting of the overlapping/shared phenotypes. The authors suggest, as a possible explanation for these overlapping phenotypes, the shared functions of imprinted genes in regulating fetal growth and their interaction in networks where disturbances in the expression of one network member may impact the expression/function of other network members. Overall, the review is comprehensive and detailed.

I have a few very minor points:

Table 1 provides a very useful overall summary. It is not clear why autism spectrum disorders are included under the column on truncal obesity for 11p15.5 region. ASD is listed for SRS but not PWS where it is a more commonly reported finding. Is it definitely known that cognitive impairment in SRS is restricted to the 7q13q32 loci which is implied by the table? It would also be helpful to have more detail on placental findings since only placental mesenchymal dysplasia is mentioned and placental findings are alluded to in figure 1. An additional column reporting these findings would be very useful for readers particularly given the function of imprinted genes in regulating placental development.

“SNVs, CNVs and UPDs belong to the spectrum of mutations which occur more or less incidentally”

Could the authors expand on what is meant by “incidentally” and it would be helpful for the non-specialised reader to have a definition of SNVs and CNVs.

Figure 1. For Temple syndrome, both loss and gain in expression from  paternal chromosome are summarised. For completeness, Silver Russel syndrome summary should include CDKN1C. Also “by affecting endocrine function and placental deficiency” would be better phrased as either “by affecting endocrine and placental function” or “by causing endocrine dysfunction and placental insufficiency”

Figure 1 legend states “The imprinting center 1 (IC1) region in 11p15 as an example for imprinting regulation and its disturbances. (LOM, loss of methylation; GOM, gain of methylation).” But this is not what is presented in the figure

Table 2. Given the theme of the review, this table might be better as Table 1. * needs to be defined

 “across the respective DMR, like KCNQ1 in 11p15.5”

Is the author referring to the long non coding RNA KCNQ1OT1 or the differentially methylated region referred to as KvDMR1?

Figure 2 does not appear to correspond with the figure legend

A few sentences could be rephrased for clarity

I am not sure that “partially transferable” is quite correct – suggest change “Since not all imprinted genes in higher mammals show the same imprinting pattern in different species, the findings from animal models are only partially transferable to the status and role of imprinted genes and clusters in human.” To “Since not all imprinted genes in higher mammals show the same imprinting pattern in different species, the findings from animal models are not always transferable to human”

“genomic imprinting plays a key role in placental physiology” would be better as “genomic imprinting plays a key role in placental development”

 “Imprinting regulation is based on the covalent modification of nucleotides (i.e. 5-methylcytosine, 5mC), posttranslational modifications of histone proteins, noncoding RNAs and chromatin confirmation. These modifications cluster at specific chromosomal regions, called differentially methylated regions (DMRs), which control the expression of imprinted”

These sentences suggest that chromatin modification “cluster” at DMRs whereas modifications can be found across imprinted domains

“However, all four molecular changes disturb the balanced expression of imprinted genes, resulting either in an increased or decreased expression.” Fundamentally, these alterations can only be “associated with” changes in gene expression in human studies

“whereas maternally expressed genes like Grb10, H19 and CDKN1C suppress it.”

I am not aware that there is experimental evidence for H19 as a direct repressor of fetal growth?

“essential prerequisites for neonatal surviving in mice”

Survival?

Change “the findings from animal models are only partially transferable to the status and role of imprinted genes and clusters in human (2)” to “are not always transferable”

Author Response

Reviewer 2:

This review provides an overview of genomic imprinting syndromes which feature fetal growth restriction principally from a clinical perspective. There is a detailed presentation of the molecular alterations reported in specific syndromes and highlighting of the overlapping/shared phenotypes. The authors suggest, as a possible explanation for these overlapping phenotypes, the shared functions of imprinted genes in regulating fetal growth and their interaction in networks where disturbances in the expression of one network member may impact the expression/function of other network members. Overall, the review is comprehensive and detailed.

I have a few very minor points:

Table 1 provides a very useful overall summary. It is not clear why autism spectrum disorders are included under the column on truncal obesity for 11p15.5 region.

ANSWER: We regret this mistake: the explanations for c and d had been mixed up.

ASD is listed for SRS but not PWS where it is a more commonly reported finding. Is it definitely known that cognitive impairment in SRS is restricted to the 7q13q32 loci which is implied by the table?

ANSWER: In fact, cognitive impairment in SRS might occur in other molecular subgroups, but it is a key feature in chromosome 7-associated cases.

It would also be helpful to have more detail on placental findings since only placental mesenchymal dysplasia is mentioned and placental findings are alluded to in figure 1. An additional column reporting these findings would be very useful for readers particularly given the function of imprinted genes in regulating placental development.

ANSWER:. The title of the “placenta” column in table 2 has been changed, and we have added a sentence about placental lesions in the introduction section, referring to the paper of Gaillot-Durand et al., 2018.

“SNVs, CNVs and UPDs belong to the spectrum of mutations which occur more or less incidentally”

Could the authors expand on what is meant by “incidentally” and it would be helpful for the non-specialised reader to have a definition of SNVs and CNVs.

ANSWER: We have modified the sentence to: “Whereas SNVs, CNVs and UPDs belong to the spectrum of naturally occurring mutations”, we hope it is better to understand now.

Figure 1. For Temple syndrome, both loss and gain in expression from  paternal chromosome are summarised. For completeness, Silver Russel syndrome summary should include CDKN1C. Also “by affecting endocrine function and placental deficiency” would be better phrased as either “by affecting endocrine and placental function” or “by causing endocrine dysfunction and placental insufficiency”

ANSWER: Done.

Figure 1 legend states “The imprinting center 1 (IC1) region in 11p15 as an example for imprinting regulation and its disturbances. (LOM, loss of methylation; GOM, gain of methylation).” But this is not what is presented in the figure

Answer: We are sorry for the mix-up of figures in the first version and have modified the order accordingly.

Table 2. Given the theme of the review, this table might be better as Table 1. * needs to be defined

 ANSWER: Done.

 “across the respective DMR, like KCNQ1 in 11p15.5”

Is the author referring to the long non coding RNA KCNQ1OT1 or the differentially methylated region referred to as KvDMR1?

ANSWER: We have modified the sentence: “…like the KCNQ1 transcript with an impact on the KCNQ1OT1:TSS-DMR in 11p15.5”.

Figure 2 does not appear to correspond with the figure legend

ANSWER: We are sorry for the mix-up of figures in the first version and have modified the order accordingly.

A few sentences could be rephrased for clarity

I am not sure that “partially transferable” is quite correct – suggest change “Since not all imprinted genes in higher mammals show the same imprinting pattern in different species, the findings from animal models are only partially transferable to the status and role of imprinted genes and clusters in human.” To “Since not all imprinted genes in higher mammals show the same imprinting pattern in different species, the findings from animal models are not always transferable to human”

ANSWER: Done.

“genomic imprinting plays a key role in placental physiology” would be better as “genomic imprinting plays a key role in placental development”

ANSWER: Done.

 “Imprinting regulation is based on the covalent modification of nucleotides (i.e. 5-methylcytosine, 5mC), posttranslational modifications of histone proteins, noncoding RNAs and chromatin confirmation. These modifications cluster at specific chromosomal regions, called differentially methylated regions (DMRs), which control the expression of imprinted”

These sentences suggest that chromatin modification “cluster” at DMRs whereas modifications can be found across imprinted domains

ANSWER: Changed.

“However, all four molecular changes disturb the balanced expression of imprinted genes, resulting either in an increased or decreased expression.” Fundamentally, these alterations can only be “associated with” changes in gene expression in human studies

ANSWER: Changed.

“whereas maternally expressed genes like Grb10, H19 and CDKN1C suppress it.”

I am not aware that there is experimental evidence for H19 as a direct repressor of fetal growth?

ANSWER: H19 has been removed.

“essential prerequisites for neonatal surviving in mice”

Survival?

ANSWER: changed.

Change “the findings from animal models are only partially transferable to the status and role of imprinted genes and clusters in human (2)” to “are not always transferable”

ANSWER: changed.